# Determination of the Popularity of Dietary Supplements Using Google Search Rankings

**DOI:** 10.3390/nu12040908

**Published:** 2020-03-26

**Authors:** Mikołaj Kamiński, Matylda Kręgielska-Narożna, Paweł Bogdański

**Affiliations:** Department of the Treatment of Obesity and Metabolic Disorders, and of Clinical Dietetics, Poznań University of Medical Sciences, Szamarzewskiego 82/84, 60-569 Poznań, Poland; matylda-kregielska@wp.pl (M.K.-N.); pawelbogdanski73@gmail.com (P.B.)

**Keywords:** google trends, dietary supplements, ranking, internet, magnesium, mineral, vitamin, protein, iron, PubMed

## Abstract

The internet provides access to information about dietary supplements and allows their easy purchase. We aimed to rank the interest of Google users in dietary supplements and to determine the changes that occurred in their popularity from 2004 to 2019. We used Google Trends to generate data over time on regional interest in dietary supplements (*n* = 200). We categorized each included supplement and calculated the interest in all topics in proportion to the relative search volume (RSV) of “lutein”. We analyzed the trends over time of all topics and categories. Globally, the topics with the highest popularity were “magnesium”, which was 23.72 times more popular than “lutein”, “protein” (15.22 times more popular), and “iron” (15.12). The categories of supplements receiving most interest were protein (9.64), mineral (5.24), and vitamin (3.47). The RSV of seven categories of topics (amino acid, bacterial, botanical, fiber, mineral, protein, and vitamin) increased over time while two categories (enzyme and fat or fatty acid) saw a drop in their RSV. Overall, 119 topics saw an increase in interest over time, 19 remained stable, and 62 saw interest in them decrease. Google Trends provides insights into e-discourse and enables analysis of the differences in popularity of certain topics across countries and over time.

## 1. Introduction

The consumption of dietary supplements is increasing globally [1]. Dietary supplements are minimally regulated and do not need a prescription. These factors, together with their broad distribution, create a positive environment for growth in their market [1]. Supplements may be sold as single-ingredient formulas or may contain multiple components [1]. The United States Office of Dietary Supplements’ Dietary Supplement Label Database collects information on the dietary supplement product labels [2]. By 2010, the database had collected over 60,000 labels with approximately a thousand new records were added each month [3]. In the National Health and Nutrition Examination Survey (NHANES) 1999–2012, approximately 52% of responders from United States consumed dietary supplements regularly [4]. However, most of the studies on the prevalence of different types of dietary supplement consumption are cross-sectional and only include a small number of respondents from specific populations [5]. There are many online commercial reports on the global dietary supplements market, but access to them is paid, and authors do not disclose the methodology of the reports. Thus, these sources might not be reliable. In our opinion, it is essential to characterize the popularity of dietary supplements across the globe because this phenomenon remains under-researched despite common nutritional supplement use. The acquired knowledge of interest in different dietary supplements may become a background for future studies. For instance, it is worthwhile to verify whether dietary supplements with low quality of scientific evidence gain the most attention.

The internet plays a pivotal role in people’s health education. The web offers immediate access to numerous sources, as well as interactions with other users. Even 90% of young French individuals perceive the internet to be a reliable source of health-related knowledge [6], enabling users to seek information on dietary supplements and also purchase the products for delivery to their home. However, as showed a Czech study, the quality of websites on dietary supplements may be low [7], and the content can be misleading [8,9,10,11]. Websites dealing with dietary supplements are often used for business purposes rather than for health promotion communication [12]. To date, the online activity of dietary supplement consumers has been poorly investigated.

Internet traffic is a source of valuable data. The investigation of internet-derived data for epidemiological purposes is termed “infodemiology” (information epidemiology) [13]. Infodemiology processes data from search engines, forums, and websites. Google is the most popular search engine globally, used by 80–90% of web users [14]. The analysis of Google queries is a novel method for predicting future infectious diseases outbreaks [15,16], assessing disease awareness [17], relative prevalence of different types of pain [18], as well as popularity of diets [19]. The analysis of searches related to dietary supplements may reveal what ingredients generate the most interest, and how this interest changes over time. Only a few studies to date have dealt with Google queries on dietary supplements. Moon et al. found that Google users are most interested in vitamin D during periods of limited sun exposure [20]. A study on bacterial supplements showed that interest in probiotics increases over time and peaks during cold months [21]. There has however been a lack of studies investigating changes in the popularity of many dietary supplements over a given period.

In this study, we aimed to analyze Google searches to address the following questions: (1) which dietary supplements generate the greatest interest among Google users globally and in different countries; and (2) what are seasonal and long-term trends in searches for dietary supplements?

## 2. Materials and Methods

### 2.1. Data Collection

This study is a retrospective infodemiological study investigating anonymous search engine statistics. This study did not require ethical committee approval. The methods we used come from previous studies [18,19].

Google Trends (GT) is an online tool for estimating the relative search volume (RSV) of searches made by Google users (https://trends.google.com/trends/). RSV is an index of search volume normalized to the number of Google users in a given period and geographical area [21]. It takes values between 0 to 100, with 100 indicating the peak of search volume (100% of popularity in given period and location) and 0 representing a complete lack of interest [22]. GT allow analysis of a chosen phrase in a selected region and in periods since January 2004, allowing comparison of up to five terms at once. In our study, we used an adjusted RSV, where a value of 100 represents the highest popularity of all the phrases being analyzed.

GT can recognize a phrase as either a “search term” or a “topic” [23]. Search terms are literal typed terms, while topics may be matched by GT when the tool recognizes phrases related to popular queries. Topics allow easy comparison of a term across languages [23]. For example, the search term “lemon” will be have the highest RSV in English speaking countries, while the topic “lemon” will include all queries associated with the matched topic in all available languages.

Due to the vast number of possible dietary supplement ingredients, we limited the final number of analyzed components to 200. We choose ingredients using the database of the National Institutes of Health’s (NIH) Dietary Supplement Label Database [2]. To our best knowledge, this database is the most detailed and regularly updated source of knowledge on dietary supplement ingredients. We examined the ingredients with the highest number of products in the “#of Products” database column. Two authors (MK and MKN) discussed each item and excluded ingredients that have no potential beneficial effects on health, which are added for their taste value, or are necessary during the production process (e.g., “Sugar”, “Fat”, “Sodium”). We simultaneously tried to match all included ingredients in GT. We typed in GT all multi-word names of dietary supplements without quotation marks. Only a few ingredients could not be matched with a topic (“citrulline malate”, “glucoamylase”, “hemicellulase”, “N-acetyl-tyrosine”), and these items were excluded from further analysis. Moving down the items in the NIH list, we searched for a corresponding topic in GT, using “health” as the default search category and “worldwide” as the region, until we obtained 200 topics. We categorized each included ingredient into one of the following groups: “amino acid”, “bacterial”, “botanical”, “chemical”, “enzyme”, “fat or fatty acid”, “fiber”, “mineral”, “protein”, or “vitamin”. The categories were derived from a group of ingredients in the Dietary Supplement Label Database. Importantly, the name “protein” may describe the category of dietary supplements or one of the included GT topics representing ingredients. The term may appear in outcomes of both types of data but represents different results. We collected the Google data from 1 January 2004 to 30 November 2019. Countries with low search volumes were excluded using GT options. We typed all the chosen topics both separately (non-adjusted data) and compared with a benchmark (adjusted data). Only two topics were compared at once. The topic “lutein” was chosen as a benchmark after an initial statistical analysis, as it had a stable long-term trend and a popularity close to the middle of the ranking, making it easy to compare with other topics. We downloaded data of interest over time and by region. All chosen GT topics, as well as the categories and the details of the search conditions and input, are presented in Appendix A, which is modified from Nuti et al. [22]. To assess whether the popularity of dietary supplements is associated with a number of scientific publications related to each nutritional supplement, we generated the list of publications from a search engine for medical literature. We chose PubMed due to: (1) free access to the search engine; (2) high popularity of this tool [24]; and (3) in contrary to Google Scholar, PubMed processes to a human-curated database [25]. For all topics, we counted the total number of publications generated by the PubMed search engine for the years 2004–2019.

### 2.2. Data Analysis

We adhered to the data processing protocol presented in an earlier analysis of data from GT [14]. We adjusted the RSVs from GT by setting all RSVs described by GT as “0” to “0.1” and all RSVs described by GT as “<1%” to “0.5”.

We used the adjusted interest over time to calculate the mean ratio of the adjusted RSV to “lutein” (Appendix A) for each topic in the study period. The topic “lutein” was represented by a value of 1.00. Furthermore, we calculated the mean ratio to the RSV of “lutein” for each category. We used Spearman’s *R* rank correlation to compare the mean ratio of the RSV of the topic “lutein” in the years 2004–2019 with the number of publications for each topic.

The adjusted data broken down by region represents the ratio between the RSV of topics and “lutein” in a specific region (Appendix A). The regions were represented by countries. The sum of both RSVs in any given region adds to 100. This allows comparison of which searches related to the topics are more often searched in a given country. We calculated the most frequent topics representing dietary supplements for all countries with a significant search volume. Because the popularity of the topics was always adjusted to the topic “lutein”, we set the RSV of “lutein” in a given country to 50. We listed the ten most popular topics in each country.

We used the unadjusted data on interest over time for time series analysis (Appendix A) for both topics and categories. For the categories, the time series was calculated as the mean RSV of all topics in a specific category. We performed the seasonal Mann–Kendall test using Kendall package, version 2.2, from R 3.6.1 (R Foundation, Vienna, Austria) to search for the presence of a significant long-term trend in time series [26]. A *p*-value below 0.05 was considered to represent a significant difference. For all significant long-term trends, we performed a univariate linear regression to determine the slope, expressed as changes in RSV per year over the study period. To analyze seasonal variation, we fitted an exponential smoothing state-space model with the Box–Cox transformation, autoregressive-moving average errors, trend, and seasonal components (TBATS) to the time trend using the forecast package, version 8.9, from R [27]. We extracted the seasonal component of the time series using seasonal decomposition of time series by loess (local polynomial regression fitting). We defined the yearly amplitude as the difference between the maximal and minimal seasonal components of the time series.

## 3. Results

Google users globally were most interested in topics representing the following categories: protein (9.64, express as a ratio with the value for “lutein”), mineral (5.24), and vitamin (3.47; Table 1). Of the 200 topics associated with dietary supplements, the most popular were “magnesium” (23.72), “protein” (15.22), “iron” (15.12), “calcium” (13.74), and “vitamin D” (12.22; Table 2 and Appendix A).

The mean proportion of a topic’s RSV to that of the topic “lutein” was positively associated with the total number of publications on the topic in PubMed in the years 2004–2019 (Rs = 0.34; *p* < 0.001; Figure 1).

A total of 149 of 250 countries and territories had low search volumes and were excluded from further analysis. Figure 2 shows the most commonly searched for topics across the 99 remaining regions. The most popular topic were “magnesium” (in 31 countries including South America, most European countries, Russia, and Turkey), “protein” (in 10 countries, mostly English speaking), “calcium” (in seven countries, including Egypt, Iran, and China), “aloe vera” and “glutathione” (in five countries), and “iron” (in four countries, including France, Israel, and Japan). Appendix A lists the ten most commonly searched for dietary supplement topics in each country.

We found that interest in seven categories of topics related to dietary supplements increased over time (amino acid, bacterial, botanical, fiber, mineral, protein, and vitamin), while interest in the fat or fatty acid category decreased over time. Interest in the enzyme and chemical categories remained stable (Figure 3 and Table 3). Interestingly, interest in all categories showed seasonal variation (Table 3). The greatest interest was generally seen in February and March, and the lowest interest was observed in December. The time-series analysis of all 200 topics related to dietary supplements is presented in Appendix A. Overall, interest in 120 topics increased over time, while 17 were stable, and 63 decreased in interest. The greatest increase over time was observed for the topics “vitamin K2” (6.11 RSV/year), “cinnamon” (5.42 RSV/year), “turmeric” (5.12 RSV/year), and “probiotic” (5.07 RSV/year). The greatest decrease in interest was noted for the topics “female ginseng” (−3.70 RSV/year), “methylsulfonylmethane” (−3.60 RSV/year), “protease” (−3.44 RSV/year), and “yohimbe” (−3.43 RSV/year). The trends of the 20 most popular topics are presented in Figure 4.

## 4. Discussion

In this retrospective infodemiological study, we ranked the global popularity of searches on topics related to dietary supplements among Google users. The analysis revealed significant differences in interest in topics across different countries, as well as over time.

### 4.1. Main Findings

NHANES is currently the largest investigation into the prevalence of supplement use. Supplement use among US citizens has remained stable from 1999 to 2012 [4], but global sales (including the US) of dietary supplements increased in the years 2011–2016, which may be due to a further increase in the prevalence of supplement use or to an increase in the number of products being taken by each consumer [28]. In the NHANES study, the most commonly used supplements were multivitamins and multiminerals [4]. In the years 2005–2012, the ten most popular of these, in order of decreasing prevalence of use, were vitamin D, vitamin C, calcium, cobalamin, vitamin E, folic acid, pyridoxine, niacin, vitamin A, and riboflavin. In the consumer survey performed by the Council for Responsible Nutrition in 2019, the ten most popular dietary supplements among US adults were multivitamins, vitamin D, vitamin C, protein, calcium, vitamin B or vitamin B complex, omega-3 fatty acids, green tea, magnesium, probiotics, iron, vitamin E, and turmeric [29]. The ranking of the popularity of dietary supplements among Google users partially reflects the data from the NHANES study and the consumer survey. The difference may be associated with the number of ingredients considered in real-world studies. Moreover, our study concerns global interest in dietary supplements, while the NHANES study is limited to the US population. Nevertheless, it seems that vitamins and minerals are both the most commonly searched for among Google users and the most commonly consumed supplements.

The popularity of the topics examined here was related to the number of PubMed publications on them. This association may be bilateral: the online discourse may be stimulated by new studies that are reported by news media; inversely, high consumption of certain supplements, and high interest in them, may motivate researchers to investigate their properties and efficacy.

The between-country differences show a high degree of diversity in interest among Google users. It is hard to clearly explain the observed popularity rank. We speculate that the greatest interest in certain dietary supplements might be associated with the geographical distribution of herbal supplements (such as aloe vera, mangosteen) or with high latitude (as with the great popularity of vitamin D in Norway). However, the great popularity of iron in France and Japan cannot be explained by the frequency of iron deficiency in those countries, both of which have a low-to-moderate prevalence of iron deficiency anemia [30]. We determined the most popular dietary supplement topic for each country and listed the ten most common of these. A valid interpretation of these rankings would require a profound knowledge of local markets, cultural circumstances, and internet discourse. We here aim to present the utility of GT and to encourage readers to interpret the rankings and make use of the results in their own investigations.

The RSV of most topics and categories increased over the study period. This suggests that general interest in dietary supplements is growing among Google users. This is not surprising, as the market for dietary supplements, as well as the number of investigations into dietary supplements, continues to increase [1].

Search trends are vulnerable to media clamor. We identified news items that could explain the peaks in Figure 4. The publication of the World Health Organization’s recommendations on salt, sodium, and potassium consumption on 1 February 2011 may explain the peak of interest in potassium in February 2011 [31]. There is a distinct peak in the interest in dietary fiber in March 2007, which could be explained by the publication on 12 March 2007 of King et al. on the effects of high fiber intake on C-reactive protein level; the article has been cited 220 times to date [32]. Similarly, a review on the positive effects of garlic on cardiovascular disease was published in April 2013 [33]. The review of Qidwai and Ashfag, which has been cited 89 times to date, may have stimulated news portals to write about garlic. The peak in search queries about iodine may be associated with the Fukushima Daiichi nuclear disaster of March 2011. The need for iodine intake to prevent thyroid cancers in individuals exposed to nuclear radiation was widely reported on [34]. The peak in searches in vitamin E in November 2004 may be related to the online publication on 10 November 2004 of a meta-analysis showing that high doses of this vitamin may increase all-cause mortality [35]. This research was widely discussed, and the media clamor may have led to an increase in interest in vitamin E.

We found that interest in all of the dietary supplement categories showed seasonality. Most Google users live in the northern hemisphere, so seasonal variation might be driven by users from Northern America, Northern Africa, Europe, and Asia [14,36]. The greatest interest was observed mainly in February and March. We hypothesize that this may be associated with the onset of seasonal outdoor activities as well as with a personal decision to lose weight or live more healthily. Search volume was lowest during December, which may be related to Christmas and winter holidays. A similar observation has been reported elsewhere [21]. However, many of the topics related to dietary supplements had the lowest interest during June, July, and August, which are the warm months in the northern hemisphere. This might be partly associated with the absence of indications (e.g., for vitamin D), the lower prevalence of common cold (for vitamin C), and the holiday period, when some users may not take time to seek information on dietary supplements. Also, a substantial number of searches over the year may be generated by scholars who are on holiday during these months, leading to a lower interest in certain topics.

### 4.2. Strengths and Practical Implications

This is the first of study to investigate the popularity of a large number of dietary supplements among Google users. The search engine statistics represent a huge number of queries from the last fifteen years. The popularity of dietary supplements differs between countries. Generally, interest in dietary supplements is on the increase among Google users. As dietary supplements do not require rigorous clinical trials, many of them lack good evidence of benefit. Moreover, the intake of dietary supplements may interact with pharmaceuticals, perhaps leading to hospitalization [37]. Unlike regulated medicines, there are less restrictions the advertisement of dietary supplements, and the internet is an attractive target for marketing campaigns. Users might in consequence be overwhelmed by targeted advertisements for dietary supplements presented as panaceas. However, Google Ads does not allow the advertising of certain dietary supplements that have been reported to be potentially harmful [38]. In confronting hyper-optimistic commercial content, health professionals should be active in online discourse and help the public understand the real benefits and harms of dietary supplements. The web community must be aware that dietary supplements cannot replace balanced diet and physical activity. Moreover, even positive results in clinical trials of efficacy should be treated with caution. Internet media may exaggerate the outcomes of some investigations to gain more attention. It is an important responsibility of professionals to present the proper interpretation of the results.

### 4.3. Limitations

The study has several limitations. First, the ingredients of dietary supplements may be searched for in a number of different contexts. We aimed to limit bias related to the non-health context of queries by choosing the “health” category in GT. However, the searches associated with biochemical components might represent scholars searching for professional information. We assume that the majority of the searches were performed by non-professional Google users. Moreover, we included topics that represent common foods, such as garlic and olives. GT has a “food and drink” category, but searches in the health category may be associated with the health properties of these foods. Second, all analyses of data from GT are affected by several limitations. First, the market share of Google differs between countries and areas. For instance, over 90% of Europeans use Google as their main search engine, but this number falls to 80–85% in the United States [36]. The data may thus be less representative of some countries. Second, GT does not provide any information regarding gender, age, socioeconomic status, occupation, lifestyle, or other user characteristics. Previous reports have suggested that young people and women were more eager to seek health-related information on the web [39,40]. In consequence, users’ age and sex might not reflect the true population distribution. However, the lack of data means that this cannot be verified.

## 5. Conclusions

Proteins, minerals, and vitamins generated the most attention among Google users in the years 2004–2019. Interest in dietary supplements generally increased from 2004 to 2019, reaching an annual peak during February and March, with the lowest interest during December. Google Trends provides insights into online discourse and allows for the analysis of differences in popularity of certain topics across countries and over time.

## Figures and Tables

**Figure 1 nutrients-12-00908-f001:**
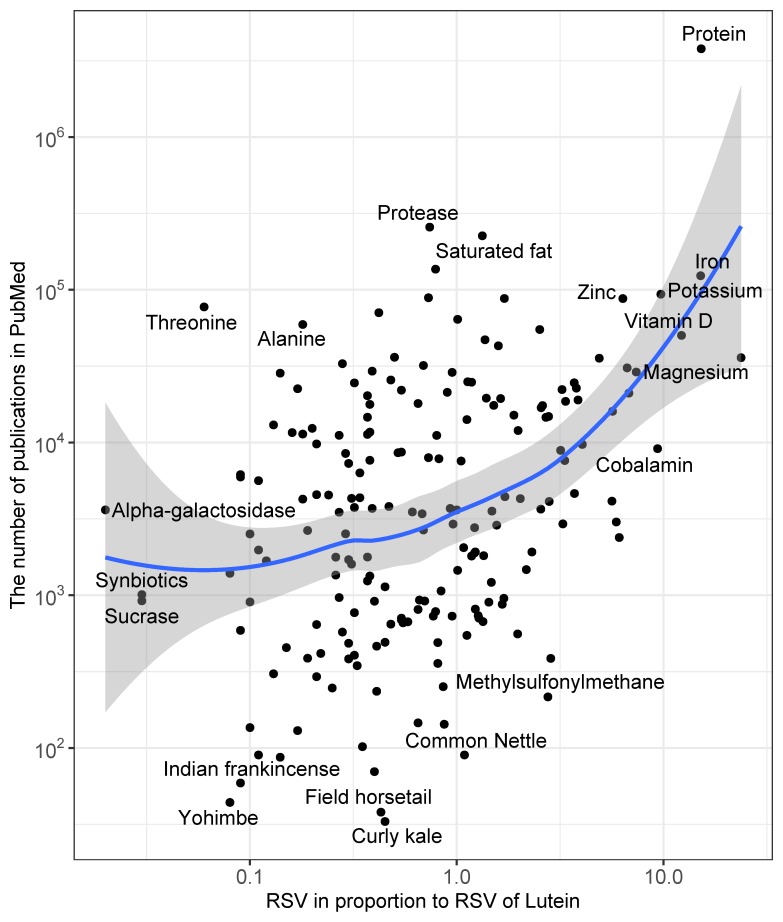
Correlation plot between the relative search volume (RSV) of each topic in proportion to the RSV of the topic “lutein” and the total number of publications in PubMed for the years 2004–2019. The plot gives the names of the topics which are outliers on the plot. The data are on an exponential scale.

**Figure 2 nutrients-12-00908-f002:**
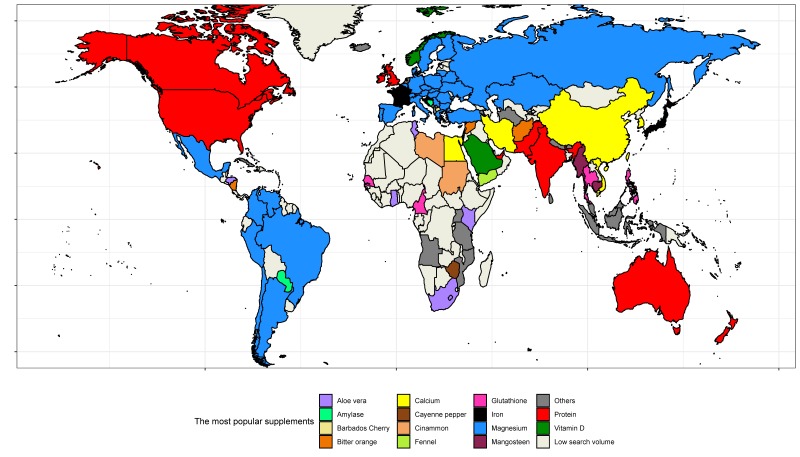
Supplements with the greatest interest by country. Countries with unique most popular supplements: Albania, boron; Angola, passion fruit; Armenia, burdock; Bhutan, cordyceps; Brunei, alfalfa; Cuba, chondroitin sulfate; Iceland, astaxanthin; Indonesia, garlic; Kosovo, iodine; Malaysia, vitamin C; Mauritius, holy basil; Montenegro, goji; Mozambique, Yohimbe; Nepal, ashwagandha; Netherlands, cobalamin; Sri Lanka, 4-aminobenzoic acid; Tanzania, nickel, Turkmenistan, glycine; Uganda, Beetroot.

**Figure 3 nutrients-12-00908-f003:**
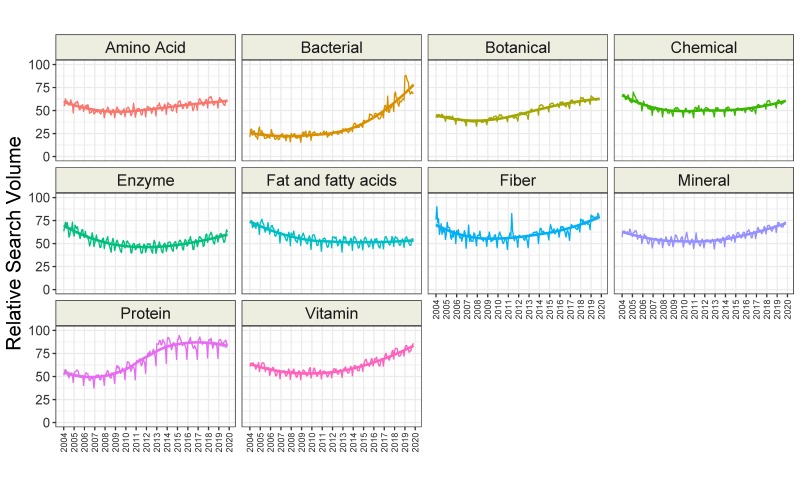
Time trends of relative search volumes of categories of dietary supplement-related topics.

**Figure 4 nutrients-12-00908-f004:**
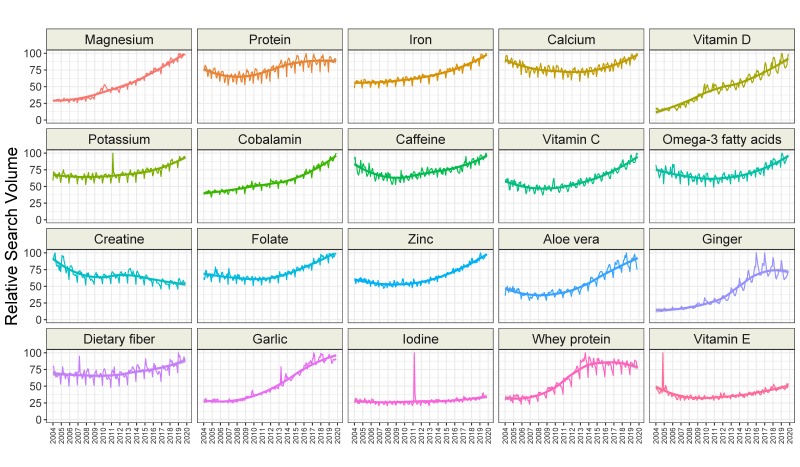
Time trends for relative search volumes of the twenty most popular dietary supplement-related topics.

**Table 1 nutrients-12-00908-t001:** Popularity of categories of supplement-related topics in proportion to “lutein” (adjusted data; relative search volume (RSV) over time).

No.	Topic	Number of Supplements in Category	Ratio of RSV to “Lutein”
1.	Protein	2	9.64
2.	Mineral	16	5.24
3.	Vitamin	17	3.47
4.	Fiber	2	3.01
5.	Chemical	19	1.76
6.	Bacterial	3	1.34
7.	Fat or fatty acid	18	1.13
8.	Botanical	91	0.97
9.	Amino acid	24	0.86
10.	Enzyme	8	0.48

**Table 2 nutrients-12-00908-t002:** Popularity of the 20 most popular topics associated with dietary supplements, as a ratio with the value for “lutein” (adjusted data; relative search volume (RSV) over time).

No.	Topic	Category	Ratio of RSV to “Lutein”
1.	Magnesium	Mineral	23.72
2.	Protein	Protein	15.22
3.	Iron	Mineral	15.12
4.	Calcium	Mineral	13.74
5.	Vitamin D	Vitamin	12.22
6.	Potassium	Mineral	9.71
7.	Cobalamin	Vitamin	9.36
8.	Caffeine	Chemical	8.06
9.	Vitamin C	Vitamin	7.38
10.	Omega-3 fatty acids	Fat or fatty acid	6.80
11.	Creatine	Chemical	6.77
12.	Folate	Vitamin	6.67
13.	Zinc	Mineral	6.36
14.	Aloe vera	Botanical	6.11
15.	Ginger	Botanical	5.91
16.	Dietary fiber	Fiber	5.67
17.	Garlic	Botanical	5.63
18.	Iodine	Mineral	4.89
19.	Whey protein	Protein	4.05
20.	Vitamin E	Vitamin	3.86

**Table 3 nutrients-12-00908-t003:** Time series analysis of categories of topics related to dietary supplements.

Topic	Seasonal Mann–Kendall Test	Slope (RSV/Year)	TBATS (Seasonality Present, Period (month))	Month with the Highest Seasonal Component (RSV)	Month with the Lowest Seasonal Component (RSV)	Seasonal Component Amplitude (RSV)
Amino acid	tau = 0.44; ***	0.47; ***	YES, 12	March (3.04)	December (−6.57)	9.61
Bacterial	tau = 0.77; ***	2.91; ***	YES, 12	February (3.97)	December (−3.73)	7.70
Botanical	tau = 0.76; ***	1.66; ***	YES, 12	March (1.88)	December (−5.36)	7.24
Chemical	tau = 0.05; 0.307	-	YES, 12	February (2.51)	December (−6.08)	8.59
Enzyme	tau = −0.06; 0.228	-	YES, 12	October (4.06)	August (−6.82)	10.88
Fat or fatty acid	tau = −0.58; ***	−1.05; ***	YES, 12	March (4.51)	December (−8.40)	12.91
Fiber	tau = 0.39; ***	0.77; ***	YES, 12	March (4.66)	December (−9.91)	14.57
Mineral	tau = 0.37; ***	0.68; ***	YES, 12	March (3.47)	December (−6.55)	10.02
Protein	tau = 0.74; ***	3.11; ***	YES, 12	March (5.07)	December (−14.02)	19.09
Vitamin	tau = 0.52; ***	1.31; ***	YES, 12	March (3.37)	December (−6.31)	9.68

*** *p*-value < 0.001. TBATS—exponential smoothing state-space model with the Box–Cox transformation, autoregressive-moving average errors, trend, and seasonal component.

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
