# Peer review of "Determination of the Popularity of Dietary Supplements Using Google Search Rankings"

_nutrients, 2020, doi:10.3390/nu12040908_

Round 1
Reviewer 1 Report
Summary
The paper titled “Determination of the popularity of dietary supplements using Google search rankings” reports on the application of Google Trends relative search volumes (RSV) as a measure of interest for dietary supplements. The authors select 200 terms, from a governmental health entity database, related to dietary supplements which are then used on the Google Trends website. Statistical analysis is performed to determine the presence of seasonal trends in the RSV behaviour. Furthermore, the authors carried a correlation analysis between the dietary supplements and PubMed articles selected with the same terms. The presented conclusion is that Google Trends provides insights into the popularity of dietary supplements, with the highest attention being given on proteins, minerals and vitamins, during the period 2004-2009, and an annual peak between February-March.
=====
Broad comments
Overall, this paper presents a study covering a large (n>2) selection of dietary supplements, however, it fails to provide an impactful research purpose; as stated by the authors, they “aim to present the utility of GT and to encourage readers to interpret the rankings and make use of the results in their own investigations” which by itself is not consistent with a research article. It lacks significant related work and it contains recurrent unsupported statements. The methodology is taken from a cited article, with additional methodology decisions unsupported. The authors briefly attempt to reason the obtained results, however, this discussion is shallow and unsupported.
Strengths
The paper presented the first study covering a large (n>2) selection of dietary supplements.
Weaknesses
- The related work is incomplete. No information is provided on past research (which does not utilise Google Trends) on dietary supplement use and popularity. The authors should also mention the many applications of Google Trends (GT) and their successes to elucidated why GT was chosen.
- The authors do not state a clear reason for this research. It is not understood what is the need and benefit of studying dietary supplement popularity. It is also unclear why seasonality and long term trends are important.
- The authors claim the methods are “inspired” by previous studies. However, in regards to citation 14 (KamiÅ„ski et al, 2019), the methodology is exactly the same; the difference is on the case study (antibiotics and probiotics vs dietary supplements). As such the authors should rephrase their statement to ensure correctness.
- It is unclear why was it necessary to categorise the supplement ingredients/topics into 10 groups. Moreover, this introduces confusion as some ingredients/topics correspond to groups as well. For example, in figure 2 supplement ingredients/topics (e.g. aloe vera) are mixed with group names (e.g. Protein).
- The authors do not mention how the search on GT was performed when using multi-word terms. This is relevant as GT interprets a multi-word in quotations (“”) as a single term while a multi-word without quotations is seen a set of n words.
- It is also unclear if the GT data is collected worldwide or by regions (line 93 vs line 102). There is also no mention of the nature of these regions until the Data Analysis section.
- The authors do not provide a reason behind the use of PubMed articles.
- It is not clear why the authors had to adjust the RSV (line 110-111) nor why they chose to compare the adjusted RSV of Lutein to the number of PubMed publications.
- The authors present the results on a regional/country RSV level yet they do not provide a profound analysis. It is unclear why these results need to be presented.
- In the Main Findings, the authors fail to address the RSV location and timeframe as a potential source of discord with the NHANES study (line 213-216).
- The authors begin to provide the reasoning for the RSV peaks, however, this is not detailed and it is limited to two news items, one of which not cited.
- It is unclear why the authors believe that frequently cited research articles might affect the search behaviour of Google users whilst stating “We assume that the majority of the searches were performed by nonprofessional Google users”. There is no supporting evidence for these assumptions.
=====
Specific comments
Unsupported statements:
- The statements on lines 33-35 are not supported by a citation.
- Citation 2 (R.E Nowak, 2010) does not support the claims for lines 35-38. Also, the authors should include a direct link to the mentioned database.
- Line 41-43, a single citation does not adequately support the author’s claims, especially given its geographic restriction.
- Lines 76-80, please cite the reference for the distinction between “search term” and “topic”. In addition, cite the reference stating that GT topics allow for comparison across languages.
- The statements in lines 222-225 are not supported by cited references.
- Line 209, “Council for Responsible Nutrition in 2019” lacks a reference.
- Line 236, “two news items” however only one is cited.
- Lines 251-263, the statements are not supported by any citation (with exception of citation 14).
Typos:
- Line 47 “only” should be “online”.
- Line 66, what is “IRB”?
- Line 76 “literally” should be “literal”.
- Line 83 “thus” should be “this”.
- Line 92, what is NIH?
- Line 178 “in observed”.
- Line 237, “Organization’s” should not be underlined.
Incoherent sentences and terms:
- Line 36 “collects information the dietary supplement” should be “collects information about dietary supplement”.
- Line 38, for coherence with the previous sentence state in which country the survey occurred.
- Line 83 “the most detailed source of knowledge on dietary supplement ingredients, and the number of records is increased regularly” should be “the most detailed and regularly updated source of knowledge on dietary supplement ingredients”.
- Line 85 “We examined the ingredients with the highest number of products”.
- Line 93 “until we had obtained 200 topics” should be “until we obtained the 200 topics”.
- The terms “ingredients” and “topics” appear to be used interchangeably.
- I suggest that the authors seek English editing services as the manuscript was difficult to follow in some sections.
Tables
- Table S1 is difficult to understand, some rows are empty.
- Table 1 and 2, the “No” column is unnecessary.
- Table 1 and 3; please leave an adequate amount of space between the column names to ensure readability. It is not possible to read table 3.
- Table 3, what do the * indicate?
Figures
- The dashes utilised in the legend of figure 2 are not consistent.
- To improve readability in figure 3 and 4, the x-axis should include a grid with vertical lines for each year between 2004 and 2019.
Author Response
Dear Reviewer,
thank you for the fast review and the comments. We hope that the new version meets your
expectations. Our responses are bolded. For our convenience, most of the revised sentences
are included in replies to our comments. All changes are highlighted.
Reviewer #1
Comments and Suggestions for Authors
Summary
The paper titled “Determination of the popularity of dietary supplements using Google search
rankings” reports on the application of Google Trends relative search volumes (RSV) as a measure
of interest for dietary supplements. The authors select 200 terms, from a governmental health entity
database, related to dietary supplements which are then used on the Google Trends website.
Statistical analysis is performed to determine the presence of seasonal trends in the RSV behaviour.
Furthermore, the authors carried a correlation analysis between the dietary supplements and
PubMed articles selected with the same terms. The presented conclusion is that Google Trends
provides insights into the popularity of dietary supplements, with the highest attention being given
on proteins, minerals and vitamins, during the period 2004-2009, and an annual peak between
February-March.
=====
Broad comments
Overall, this paper presents a study covering a large (n>2) selection of dietary supplements,
however, it fails to provide an impactful research purpose; as stated by the authors, they “aim to
present the utility of GT and to encourage readers to interpret the rankings and make use of the
results in their own investigations” which by itself is not consistent with a research article. It lacks
significant related work and it contains recurrent unsupported statements. The methodology is taken
from a cited article, with additional methodology decisions unsupported. The authors briefly
attempt to reason the obtained results, however, this discussion is shallow and unsupported.
RE: Dear Reviewer, thank you for this detailed revision. We read your comments carefully
and address all of them.
Strengths
The paper presented the first study covering a large (n>2) selection of dietary supplements.
Weaknesses
- The related work is incomplete. No information is provided on past research (which does not utilise
Google Trends) on dietary supplement use and popularity.
- RE: Most of the surveys on dietary supplement use are limited in the number of participants,
design (cross-sectional), number of analyzed products as well as investigated population (e.g.,
only students, not general population). Therefore, we did not expand the introduction with
unnecessary references. To our best knowledge, our study investigates the highest number of
different dietary supplements at once.
The authors should also mention the many applications of Google Trends (GT) and their successes to
elucidated why GT was chosen.
RE: We added to the introduction:
„The analysis of Google queries is a novel method for predicting future infectious diseases
outbreaks [15,16], assessing disease awareness [17], relative prevalence of different types
of pain [18] as well as popularity of diets [19].”
- The authors do not state a clear reason for this research. It is not understood what is the need and
benefit of studying dietary supplement popularity. It is also unclear why seasonality and long term
trends are important.
RE: To address this question , we added to the Introduction:
„However, most of the studies on the prevalence of different types of dietary supplements
consumption are cross-sectional, include a small number of respondents from specific
populations [5]. There are many online commercial reports on the global dietary
supplements market, but access to them is paid, and authors do not disclose the
methodology of the reports; thus, these sources might not be reliable. In our opinion, it is
essential to characterize the popularity of dietary supplements across the globe because
this phenomenon remains under-researched despite common nutritional supplement use.
The acquired knowledge of interest in different dietary supplements may become a
background for future studies. For instance, it is worthwhile to verify whether dietary
supplements with low quality of scientific evidence gain the most attention. ”
- The authors claim the methods are “inspired” by previous studies. However, in regards to citation
14 (Kamiński et al, 2019), the methodology is exactly the same; the difference is on the case study
(antibiotics and probiotics vs dietary supplements). As such the authors should rephrase their
statement to ensure correctness.
RE: Thank you for this comment. We rephrased the sentence:
„The methods we use comes from previous studies [18,19].”
- It is unclear why was it necessary to categorise the supplement ingredients/topics into 10 groups.
RE: It was done to simplify the interpretation of the data. Results of 200 topics are vast in aimed
to include in the manuscript only essential results. Others are presented in the supplementary
material.
Moreover, this introduces confusion as some ingredients/topics correspond to groups as well. For
example, in figure 2 supplement ingredients/topics (e.g. aloe vera) are mixed with group names
(e.g. Protein).
RE: Thank you for this comment. We added to the Methods section:
„Importantly, the name “Protein” may describe the category of dietary supplements or one
of the included GT topic representing ingredient. The term may appear in outcomes of
both types of data but represents different results.”
- The authors do not mention how the search on GT was performed when using multi-word terms.
This is relevant as GT interprets a multi-word in quotations (“”) as a single term while a multi-word
without quotations is seen a set of n words.
RE: We added to the Methods section:
„We typed in GT all multi-word names of dietary supplements without quotation marks.”
- It is also unclear if the GT data is collected worldwide or by regions (line 93 vs line 102). There is
also no mention of the nature of these regions until the Data Analysis section.
RE: This refers to two different processes. The data was collected for region worldwide, but the
generated results may be downloaded as „trends over time” and „by each region”.
See an example from Google Trends: https://trends.google.com/trends/explore?date=2014-01-
01%202019-08-31&q=%2Fm%2F0hkwr,%2Fm%2F06xwy1
We changed the sentence in (previous) line 102:
„We downloaded data of interest over time and by region.”
- The authors do not provide a reason behind the use of PubMed articles.
RE: Thank you for this comment. We added to the Methods section:
„To assess whether the popularity of dietary supplements is associated with a number of
scientific publications related to each nutritional supplement, we generated the list of
publications from a search engine for medical literature. We chose PubMed due to 1) free
access to the search engine, 2) high popularity of this tool [24], and 3) in contrary to Google
Scholar, the Pubmed processes to a human-curated database [25]. For all topics, we
counted the total number of publications generated by the PubMed search engine for the
years 2004-2019.”
- It is not clear why the authors had to adjust the RSV (line 110-111) nor why they chose to compare
the adjusted RSV of Lutein to the number of PubMed publications.
RE: The adjusted data allow to rank the popularity of the analyzed topic. Lutein is only a
benchmark. Therefore, we may analyze the correlation between the relative popularity of 200
dietary supplements and the number of publications generated by PubMed.
- The authors present the results on a regional/country RSV level yet they do not provide a
profound analysis. It is unclear why these results need to be presented.
RE: These results are additional and may picture heterogeneity of the interest in dietary
supplements in different countries. We included the detailed top 10 rankings for each
analyzed country in the supplementary file. In our opinion, the readers may be interested to
see the most popular dietary supplements in the country of their origin.
- In the Main Findings, the authors fail to address the RSV location and timeframe as a
potential source of discord with the NHANES study (line 213-216).
RE: Thank you for this comment. We added to the paragraph:
„Moreover, our study concerns global interest in dietary supplements, while the NHANES
study is limited to the US population.”
- The authors begin to provide the reasoning for the RSV peaks, however, this is not detailed
and it is limited to two news items, one of which not cited.
RE: This problem is also mentioned below. The paragraph was rephrased and we removed
„two”:
„Search trends are vulnerable to media clamor. We identified news items that could explain the peaks in
Figure 4. The publication of the World Health Organization’s recommendations on salt, sodium, and
potassium consumption on 1 February 2011 may explain the peak of interest in potassium in February 2011
[25]. There is a distinct peak in the interest in dietary fiber in March 2007, which could be explained by the
publication on 12 March 2007 of King et al. on the effects of high fiber intake on C-reactive protein level; the
article has been cited 220 times to date [26]. Similarly, a review on the positive effects of garlic on
cardiovascular disease was published in April 2013 [27]. The review of Qidwai and Ashfag, which has been
cited 89 times to date, may have stimulated news portals to write about garlic. The peak in search queries
about iodine may be associated with the Fukushima Daiichi nuclear disaster of March 2011. The need for
iodine intake to prevent thyroid cancers in individuals exposed to nuclear radiation was widely reported on
[28]. The peak in searches in vitamin E in November 2004 may be related to the online publication on 10
November 2004 of a meta-analysis showing that high doses of this vitamin may increase all-cause mortality
[29]. This research was widely discussed, and the media clamor may have led to an increase in interest in
vitamin E. ”
- It is unclear why the authors believe that frequently cited research articles might affect the
search behaviour of Google users whilst stating “We assume that the majority of the searches were
performed by nonprofessional Google users”. There is no supporting evidence for these
assumptions.
RE: There is no supporting evidence, and this is only the assumption. Since we analyzed
topics representing dietary supplements that are commonly consumed, we speculate that
nonprofessionals did most of the queries in Google. Professionals may be more interested in
scientific search engines.
=====
Specific comments
Unsupported statements:
- The statements on lines 33-35 are not supported by a citation.
RE: Corrected. These statements also come from review of Binns et al.: (C.W. Binns, M.K.
Lee, A.H. Lee, Problems and prospects: public health regulation of dietary supplements,
Annu. Rev. Public Health 39 (2018) 403–420. https://doi.org/10.1146/annurev-publhealth-
040617-013638.)
- Citation 2 (R.E Nowak, 2010) does not support the claims for lines 35-38. Also, the authors should
include a direct link to the mentioned database.
RE: We cited the following reference: „Dietary Supplement Label Database. Alphabetical
Listing of Ingredients, (n.d.). https://dsld.nlm.nih.gov/dsld/lstIngredients.jsp (accessed
December 7, 2019).” (primarily, this references was used in Methods section.
- Line 41-43, a single citation does not adequately support the author’s claims, especially given its
geographic restriction.
RE: Currently, the sentence was rewritten: „Even 90% of young French individuals perceive
the internet to be a reliable source of health-related knowledge [5], enabling users to seek
information on dietary supplements and also purchase the products for delivery to their
home.” . The study of Beck et al. (reference no 6) was limited do young French adults, and we
cited this paper to show an example of how high might be trusted the internet as a reliable
source. The second part of this sentence is common knowledge, and in our opinion, it did not
require citation.
- Lines 76-80, please cite the reference for the distinction between “search term” and “topic”. In
addition, cite the reference stating that GT topics allow for comparison across languages.
RE: We added citation from Google Trends Help:
Compare Trends search terms, (n.d.). https://support.google.com/trends/answer/4359550?
hl=en (accessed March 15, 2020).
- The statements in lines 222-225 are not supported by cited references.
RE: These statements are our suppositions, we cannot provide a references. We edited this
fragment:
„We speculate that the greatest interest in certain dietary supplements might be associated
with the geographical distribution of herbal supplements (such as aloe vera, mangosteen)
or with high latitude (as with the great popularity of vitamin D in Norway).”
- Line 209, “Council for Responsible Nutrition in 2019” lacks a reference.
RE: The reference was at the very end of this sentence:
„In the consumer survey performed by the Council for Responsible Nutrition in 2019, the ten most popular
dietary supplements among US adults were multivitamins, vitamin D, vitamin C, protein, calcium, vitamin
B or vitamin B complex, omega-3 fatty acids, green tea, magnesium, probiotics, iron, vitamin E, and turmeric
[22].”
- Dietary supplement use reaches all-time high: Available-for-purchase consumer survey reaffirms the vital role
supplementation plays in the lives of most Americans, (2019). https://www.crnusa.org/newsroom/dietarysupplement-
use-reaches-all-time-high-available-purchase-consumer-survey-reaffirms (accessed January 14, 2020).
- Line 236, “two news items” however only one is cited.
RE: Thank you for this comment. We edited this sentence, and removed „two”:
„We identified news items that could explain the peaks in Figure 4.”
- Lines 251-263, the statements are not supported by any citation (with exception of citation 14).
RE: This paragraph presents our hypotheses on possible explanations. Citations supported
the statement on the distribution of Google users across two hemispheres on search engine
market shares across the world:
„We found that interest in all of the dietary supplements categories showed seasonality. Most Google users
live in the northern hemisphere, so seasonal variation might be driven by users from Northern America,
Northern Africa, Europe, and Asia [13,30].”
Typos:
- Line 47 “only” should be “online”.
RE: Corrected
- Line 66, what is “IRB”?
RE: Corrected:
„This study did not require ethical comittee approval.”
- Line 76 “literally” should be “literal”.
RE: Corrected
- Line 83 “thus” should be “this”.
RE: Corrected
- Line 92, what is NIH?
RE: Corrected:
The sentence:
„ We choose ingredients using the database of the National Institutes of Health’s (NIH)
Dietary Supplement Label Database [18].” missed an abbreviation
- Line 178 “in observed”.
RE: Corrected:
„was observed”
- Line 237, “Organization’s” should not be underlined.
RE: Corrected
Incoherent sentences and terms:
- Line 36 “collects information the dietary supplement” should be “collects information about dietary
supplement”.
RE: Corrected
„collects information on the dietary supplement product labels”
- Line 38, for coherence with the previous sentence state in which country the survey occurred.
RE: Corrected:
„In the National Health and Nutrition Examination Survey (NHANES) 1999-2012,
approximately 52% of responders from United States consumed dietary supplements
regularly [3].”
- Line 83 “the most detailed source of knowledge on dietary supplement ingredients, and the number
of records is increased regularly” should be “the most detailed and regularly updated source of
knowledge on dietary supplement ingredients”.
RE: Thank you for this comment, corrected.
- Line 85 “We examined the ingredients with the highest number of products”.
RE: Currently, the sentence: „We examined the ingredients with the highest number of
products in the “#of Products” database column.” indicate which column from a cited source
we choose (Dietary Supplement Label Database. Alphabetical Listing of Ingredients, (n.d.).
https://dsld.nlm.nih.gov/dsld/lstIngredients.jsp (accessed December 7, 2019).)
- Line 93 “until we had obtained 200 topics” should be “until we obtained the 200 topics”.
RE: Corrected
- The terms “ingredients” and “topics” appear to be used interchangeably.
RE: Because the name of the topics were based on NIH ingredient list
- I suggest that the authors seek English editing services as the manuscript was difficult to follow in
some sections.
RE: This study was previously proofread by professional editing service. We corrected all the
errors noticed by the reviewers.
Tables
- Table S1 is difficult to understand, some rows are empty.
RE: We made a fault, and Table S1 lacked section b) with all topics. Table S1 section a) is a
standard protocol of Nuti et al. (https://doi.org/10.1371/journal.pone.0109583). All bolded
phrases represent headings and subheadings of the protocol.
- Table 1 and 2, the “No” column is unnecessary.
RE: We are aware that without this column name, the table is readable. However, it is a good
custom to fill all column names; thus, we decided to leave „No.” above the column with the
rank number.
- Table 1 and 3; please leave an adequate amount of space between the column names to ensure
readability. It is not possible to read table 3.
RE: Thank you for this comment. We corrected the column names.
- Table 3, what do the * indicate?
RE: By mistake, we did not included the legend under the Table 3:
„*** p-value < 0.001”
Figures
- The dashes utilised in the legend of figure 2 are not consistent.
RE: Dear Reviewer, we do not understand this comment. Figure 2 is a world map. What
dashes in the legend do you mean? Could you provide an example that shall we follow?
- To improve readability in figure 3 and 4, the x-axis should include a grid with vertical lines for each
year between 2004 and 2019.
RE: We edited both figures.
Reviewer 2 Report
This is likely a topic of interest for those interested in nutrition research and marketing.
My biggest concern was the reports on seasonality when analysing data from the northern and southern hemispheres. This should be analysed separately to account for the differences in seasons eg. Vit D might be of more interest in June-July in the Southern hemisphere. Analysing together may mask stronger seasonal effects in the 2 hemispheres.
There are some typos that need fixing. Line 36. information ON the...
Line 46. Sentence starting with "Despite" should be reworded as it is unclear
Line 83. To the best of our knowledge THIS database..
Did you search terms with different spelling eg. Fiber (American) vs Fibre (British English)?
Broad statement have been made about studies performed in a particular country. eg. Ref [4] Beck et al was a study examining French adults and you should clarify this in line 42.
Similarly ref [5] which is about supplements in the Czech Republic. You should clarify this in line 44.
Author Response
Dear Reviewer,
thank you for the fast review and the comments. We hope that the new version meets your
expectations. Our responses are bolded. For our convenience, most of the revised sentences
are included in replies to our comments. All changes are highlighted.
Reviewer #2
This is likely a topic of interest for those interested in nutrition research and marketing.
My biggest concern was the reports on seasonality when analysing data from the northern and
southern hemispheres. This should be analysed separately to account for the differences in seasons
- Vit D might be of more interest in June-July in the Southern hemisphere. Analysing together
may mask stronger seasonal effects in the 2 hemispheres.
RE:
Seasonality of searches on Vitamin D was previously analyzed by Moon et al. They compared
seasonality in different countries in both hemispheres.
- R.J. Moon, E.M. Curtis, J.H. Davies, C. Cooper, N.C. Harvey, Seasonal variation in Internet searches for vitamin
D, Arch. Osteoporos. 12 (2017) 28. https://doi.org/10.1007/s11657-017-0322-7.
We aimed to analyze global seasonal variations. If you wish, we can make an additional
analysis comparing seasonal variation of all (or some) topics in countries representing both
hemispheres, but it will takes more time.
There are some typos that need fixing. Line 36. information ON the...
RE: Corrected
Line 46. Sentence starting with "Despite" should be reworded as it is unclear
RE: We rewritten the sentence:
„To date, the online activity of dietary supplement consumers has been poorly investigated. ”
Line 83. To the best of our knowledge THIS database..
RE: Corrected
Did you search terms with different spelling eg. Fiber (American) vs Fibre (British English)?
RE: Importantly, the topics include translation of matched search terms as well as synonyms
and phrases, including matched search terms e.g., „fiber purchase”, „fiber properties,” etc.
The search term „Fibre” is not matched as the topic „Dietary fiber” by GT, but „Fibre
dietary” can be matched as the topic „Dietary fiber”. Moreover, „fibre” is recognized as a
synonym of the keyword „fiber” in Google Ads (see screenshot below). Therefore, it may be
assumed that the topic „Dietary fiber” includes a vast number of search terms.
Broad statement have been made about studies performed in a particular country. eg. Ref [4] Beck
et al was a study examining French adults and you should clarify this in line 42.
RE: We rewritten the sentence:
“Even 90% of young French individuals perceive the internet to be a reliable source of health-related
knowledge”
Similarly ref [5] which is about supplements in the Czech Republic. You should clarify this in line
44.
RE: We rewritten the sentence:
“However, as showed a Czech study, the quality of websites on dietary supplements may be low [7],
and the content can be misleading [8-11].”
Reviewer 3 Report
The research questions in this paper do not require an advanced analysis. Perhaps, it would be worth adding a research problem, e.g.
How does advertising affect the popularity of dietary supplements?
As for the visual aspect (the layout) - there should be a larger gap between heading 2 and heading 3 in table 1.
As for the table 3, the column headings should be assigned to the data in these columns.
Author Response
Dear Reviewer,
thank you for the fast review and the comments. We hope that the new version meets your
expectations. Our responses are bolded. For our convenience, most of the revised sentences
are included in replies to our comments. All changes are highlighted.
Reviewer #3
The research questions in this paper do not require an advanced analysis. Perhaps, it would be worth
adding a research problem, e.g.
How does advertising affect the popularity of dietary supplements?
RE:
We raised this matter in the discussion:
“Unlike regulated medicines, there are less restrictions the advertisement of dietary supplements,
and the internet is an attractive target for marketing campaigns. Users might in consequence be
overwhelmed by targeted advertisements for dietary supplements presented as panaceas. However,
Google Ads does not allow the advertising of certain dietary supplements that have been reported to
be potentially harmful [30]. In confronting hyperoptimistic commercial content, health
professionals should be active in online discourse and help the public understand the real benefits
and harms of dietary supplements. The web community must be aware that dietary supplements
cannot replace balanced diet and physical activity. Moreover, even positive results in clinical trials
of efficacy should be treated with caution. Internet media may exaggerate the outcomes of some
investigations to gain more attention.”
In our opinion, we cannot assess how does advertising affects the popularity of dietary
supplements using data from Google Trends. To address this question, it is worth considering
a survey study. The questionnaire should investigate how e-adverts, bloggers opinion on
dietary supplements, etc. affects purchases of the respondents.
However, the most valuable will be the data on e-advertisements: name of advertised dietary
supplement, clicks on e-ad (conversion), details on region and language where these ads are
displayed etc.
As for the visual aspect (the layout) - there should be a larger gap between heading 2 and heading 3
in table 1.
RE:Corrected
As for the table 3, the column headings should be assigned to the data in these columns.
RE:Corrected
Reviewer 4 Report
This manuscript uses Google Trends to rank the interest of Google users in dietary supplements over a 15 year period (2004-2019), both globally and in different countries. It also looks at seasonal and long-term trends in interest of Google users. The manuscript, along with it’s tables and figures, is easy to read and follow, interesting and informative although being somewhat of a scientific “light-weight”. That said, it gives practical information and the limitations of the study are quite well summarized in the Discussion. Also, the discussion on different news items that may explain peaks in interest of selected dietary supplements is good.
Specific comments:
Line 36: “information the dietary” a word is missing, might be changed to “information off the dietary”.
Line 38: “records are added” should be changed to “records were added” since this is based on 10 year old data.
Line 39: would be valuable to see the years included in the NHANES study, e.g. “(NHANES) 1999-2012, approximately (...)”
Line 40: “patient’s health education” might be expanded to include the general public (without disease), e.g. “The internet plays a pivotal role in people’s health education”.
Line 41: Might add: As many of 90% of individuals “in certain groups” perceive the internet (...), since the study referred to includes only young adults, who might possibly be the group most positive about the reliability of the internet.
Line 47: “the only activity” should be changed to “the online activity”
Line 56: “insolation” might be changed to “sunlight” or “sun exposure”, more commonly used.
Line 60: “locally” might be changed to “in different countries” as locally could be understood as locally for you (Poland).
Line 83: “thus database” should be changed to “this database”
Line 167 (Table S4): Table S4 only includes 61 countries, not the 101 countries you would expect.
Figure 2: Some of the colours, e.g. the green for amylase and vitamin D, are quite similar and hard to distinguish between them.
Lines 176 and 177: “Table 4” should be “Table 3”, there is no Table 4.
Table 3: What does the “***” stand for? P-value below (?)
Line 237: Do not underline “Organization’s”
Author Response
Dear Reviewer,
thank you for the fast review and the comments. We hope that the new version meets your
expectations. Our responses are bolded. For our convenience, most of the revised sentences
are included in replies to our comments. All changes are highlighted.
Reviewer #4
This manuscript uses Google Trends to rank the interest of Google users in dietary supplements
over a 15 year period (2004-2019), both globally and in different countries. It also looks at seasonal
and long-term trends in interest of Google users. The manuscript, along with it’s tables and figures,
is easy to read and follow, interesting and informative although being somewhat of a scientific
“light-weight”. That said, it gives practical information and the limitations of the study are quite
well summarized in the Discussion. Also, the discussion on different news items that may explain
peaks in interest of selected dietary supplements is good.
RE: Thank you for the appreciation.
Specific comments:
Line 36: “information the dietary” a word is missing, might be changed to “information off the
dietary”.
RE: Corrected.
Line 38: “records are added” should be changed to “records were added” since this is based on 10
year old data.
RE: Corrected.
Line 39: would be valuable to see the years included in the NHANES study, e.g. “(NHANES) 1999-
2012, approximately (...)”
RE: Corrected.
Line 40: “patient’s health education” might be expanded to include the general public (without
disease), e.g. “The internet plays a pivotal role in people’s health education”.
RE: Good point. Thank you for this hint. Corrected.
Line 41: Might add: As many of 90% of individuals “in certain groups” perceive the internet (...),
since the study referred to includes only young adults, who might possibly be the group most
positive about the reliability of the internet.
RE: We edited the sentence for:
„Even 90% of young French individuals perceive the internet to be a reliable source of healthrelated
knowledge [4],”
Line 47: “the only activity” should be changed to “the online activity”
RE: Corrected.
Line 56: “insolation” might be changed to “sunlight” or “sun exposure”, more commonly used.
RE: We exchanged „insolation” for „sun exposure”
Line 60: “locally” might be changed to “in different countries” as locally could be understood as
locally for you (Poland).
RE: Corrected.
Line 83: “thus database” should be changed to “this database”
RE: Corrected.
Line 167 (Table S4): Table S4 only includes 61 countries, not the 101 countries you would expect.
RE: Dear Reviewer, I made a fault in calculation. In summary we ranked popularity of
dietary supplements in 99 countries. I also corrected the Table S4. Now it includes 99
countries.
Figure 2: Some of the colours, e.g. the green for amylase and vitamin D, are quite similar and hard
to distinguish between them.
RE: Due to many variables we had many troubles with visualization on Figure 1.
Nevertheless, all dietary supplements have different colours: amylase has turquise, and
vitamin D dark green.
If you have any preposition to change the color, please do not hesitate to write a comment.
Lines 176 and 177: “Table 4” should be “Table 3”, there is no Table 4.
RE: Corrected.
Table 3: What does the “***” stand for? P-value below (?)
RE: Corrected. We added caption below the Table: „*** p-value < 0.001”
Line 237: Do not underline “Organization’s”
RE: Corrected.